# Gas hydrate dissociation off Svalbard induced by isostatic rebound rather than global warming

Klaus Wallmann[1], M. Riedel [1], W.L. Hong [2,3], H. Patton [3], A. Hubbard [3,4], T. Pape[5], C.W. Hsu[5], C. Schmidt[1], J.E. Johnson[6], M.E. Torres[7], K. Andreassen[3], C. Berndt [1] & G. Bohrmann [5]

Methane seepage from the upper continental slopes of Western Svalbard has previously been attributed to gas hydrate dissociation induced by anthropogenic warming of ambient bottom waters. Here we show that sediment cores drilled off Prins Karls Foreland contain freshwater from dissociating hydrates. However, our modeling indicates that the observed pore water freshening began around 8 ka BP when the rate of isostatic uplift outpaced eustatic sea-level rise. The resultant local shallowing and lowering of hydrostatic pressure forced gas hydrate dissociation and dissolved chloride depletions consistent with our geochemical analysis. Hence, we propose that hydrate dissociation was triggered by postglacial isostatic rebound rather than anthropogenic warming. Furthermore, we show that methane fluxes from dissociating hydrates were considerably smaller than present methane seepage rates implying that gas hydrates were not a major source of methane to the oceans, but rather acted as a dynamic seal, regulating methane release from deep geological reservoirs.

[1] GEOMAR Helmholtz Centre for Ocean Research Kiel, Wischhofstr. 1-3, Kiel 24148, Germany. [2] Geological Survey of Norway, N-7022 Trondheim, Norway. [3] CAGE Centre for Arctic Gas Hydrate Research, Environment and Climate, Department of Geosciences, UiT—The Arctic University of Norway, Tromsø N-9037, Norway. [4] Department of Geography & Earth Science, Aberystwyth University, Wales SY23 3DB, UK. [5] MARUM—Center for Marine Environmental Sciences and Department of Geosciences, University of Bremen, Klagenfurter Str., Bremen 28359, Germany. [6] Department of Earth Sciences, University of New Hampshire, 56 College Rd., Durham, NH 03824-3589, USA. [7] College of Oceanic and Atmospheric Sciences, Oregon State University, 104 Ocean Admin Building, Corvallis, OR 97331–5503, USA. Correspondence and requests for materials should be addressed to K.W. (email: kwallmann@geomar.de)

Vast amounts of methane are bound in gas hydrates that accumulate in seafloor sediments across continental margins. These ice-like solids are stable under high pressure/ low temperature conditions but dissociate under ocean warming or relative sea-level lowering. The global gas hydrate inventory totals some 1000 billion metric tons of carbon[1], the decomposition of which would affect carbon cycling and climate on the global scale[2–4]. Hence, hydrate dissociation has been invoked to explain many observations, such as the Paleocene-Eocene Thermal Maximum[2] and the rapid postglacial increase in atmospheric methane[5]. Whereas seafloor methane emissions and the associated formation of $^{13}$C-depleted carbonates have been ascribed to hydrate dissociation[6–8], direct evidence for the latter process is still conspicuously lacking. Nevertheless, it is argued that a positive feedback associated with methane release from widespread hydrate dissociation could amplify future global warming[4].

Observed methane seepage from the upper continental slope of northwestern Svalbard at ~ 400 m water depth has been attributed to gas hydrate dissociation induced by warming of ambient bottom waters and postulated as the onset stage of this future trend[7]. Numerical modeling studies support this hypothesis since numerous seepage sites are located at the up-dip limit of the gas hydrate stability zone where a moderate rise in ambient bottom water temperature would induce hydrate decomposition[9]. However, gas hydrates have never been sampled from the upper slope margin off northwest Svalbard, and dating of authigenic carbonates associated with the methane seeps reveals that seepage has been active for at least 3000 years[8]. Moreover, methane seepage is also known to prevail at depths shallower than the hydrate stability zone[10, 11] and a hydrate-bearing seep area south of Svalbard shows limited influence from short-term ocean warming[12]. Hence, methane seepage from the seafloor may not originate from dissociating gas hydrates but from free gas that migrates to the seafloor along high-permeability stratigraphic or structural conduits[10].

Here, we present the first geochemical data that unequivocally confirm gas hydrate dissociation in sediments cored off Western Svalbard. We find remnant freshwater from hydrate dissociation that was formed over the last 8000 years when isostatic rebound induced by the deglaciation of the Barents Sea ice sheet outpaced eustatic sea-level rise. Furthermore, we find that seafloor methane seepage subsequently increased because the permeability of sediments was enhanced by the decay of hydrates that previously clogged the pore space, thereby enhancing methane release from underlying geological reservoirs.

## Results

**Sampling.** During R/V MARIA S. MERIAN cruise MSM57 in August 2016, sediment cores were recovered using the MARUM-MeBo70 drill-rig and a conventional gravity corer (GC) at the upper slope off northwestern Svalbard where numerous gas flares were previously identified (Fig. 1)[11]. A micro-temperature logger (MTL, Antares type 1854) was modified to fit into the core pilot tube to measure in situ formation temperature during MeBo deployments. The cores were analyzed for porosity while dissolved chloride and sulfate concentrations were determined in pore fluids separated from the bulk sediment (as outlined in the methods section).

**Sediment and pore water composition.** Sediments in the recovered cores are mixed hemipelagic to glaciomarine deposits composed of a wide range of grain sizes from clay to sand with variable amounts of gravel to pebble-sized rocks. They were deposited by ice-rafting and/or as glacial debris flows associated with nearby trough-mouth-fan deposition[13] and bottom current activity[14] on the upper slope during the Late Pleistocene. Our measurements indicate a down-core temperature increase associated with a geothermal gradient of 45–50 °C km$^{-1}$ and a general decline in porosity, dissolved chloride and sulfate with sediment depth (Fig. 2). Porosity profiles reflect compaction and random grain size variations with low-porosity sections dominated by sand/boulder intervals and high-porosity layers associated with significant clay/silt contents. Sulfate is removed from the pore water by microbial sulfate reduction and the anaerobic oxidation of methane (AOM). Elevated sulfate concentrations below 5 m sediment depth detected in cores GeoB21632-1 and GeoB21639-1 are probably artifacts caused by the intrusion of sulfate-bearing seawater that was employed as drilling fluid and penetrated into

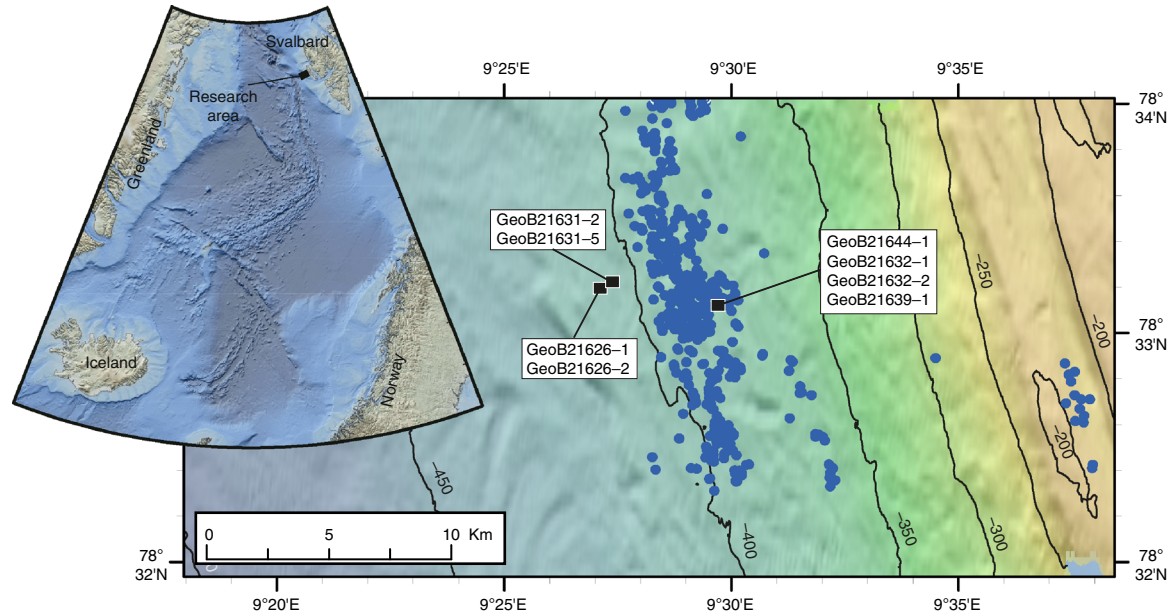

**Fig. 1** Location of coring sites and gas flares. Gas flares (blue dots) were identified during a previous cruise[11]. Locations are listed in Supplementary Table 1

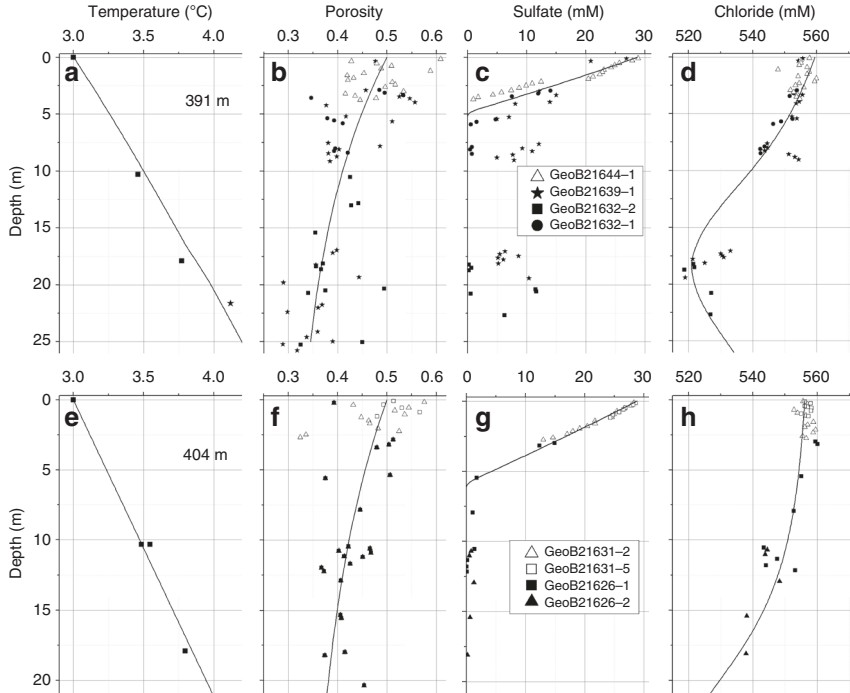

**Fig. 2** In situ temperature, porosity, and pore water composition. Symbols indicate observations and lines represent best fitting model results. Chloride concentrations were corrected to account for seawater intrusion during the drilling process (methods section). **a** In situ temperature at 391 m water depth. **b** Porosity at 391 m water depth. **c** Dissolved sulfate in pore fluids at 391 m water depth. **d** Dissolved chloride in pore fluids at 391 m water depth. **e** In situ temperature at 404 m water depth. **f** Porosity at 404 m water depth. **g** Dissolved sulfate in pore fluids at 404 m water depth. **h** Dissolved chloride in pore fluids at 404 m water depth

permeable sediment layers. Dissolved inorganic carbon is strongly depleted in $^{13}C$ at the base of the sulfate-bearing zone (Supplementary Fig. 1). The significant negative $\delta^{13}C$ values (−40‰) are driven by AOM[15] rather than the degradation of organic matter[16]. The down-core increase towards positive $\delta^{13}C$-DIC values (up to + 17‰) may reflect active methanogenesis via $CO_2$ reduction leaving behind a residual dissolved inorganic carbon (DIC) pool enriched in $^{13}C$[15]. The isotopic composition of methane at the base of the cores (−53‰) is characteristic for biogenic gas containing a small but significant admixture of thermogenic methane from deeper sources[17]. It is similar to the isotopic composition of gas seeping from the seabed[11, 16] and gas bound in methane hydrates sampled at 890 m water depth[18].

Dissolved chloride decreases significantly with sediment depth (Fig. 2). None of the drill cores contained gas hydrates and measurements with an infrared camera conducted within 1 h after core retrieval showed no negative temperature anomaly indicative for endothermic gas hydrate dissociation[19]. The in situ temperature measurements clearly show that methane hydrate was not stable in the cores taken at 391 m water depth, whereas at 404 m only the uppermost sediment section was located within the gas hydrate stability zone during the time of sampling (Fig. 3)[20]. Hence, we conclude that the observed chloride depletion is not an artifact caused by gas hydrate dissociation upon core retrieval but rather indicates in situ admixture of freshwater. The isotopic composition ($\delta^{18}O$, $\delta^{2}H$) of the pore fluids and their lithium and boron content (Supplementary Figs. 2 and 3) indicate that the freshwater indeed originates from gas hydrate dissociation[21] that occurred when temperatures increased to their present level and/or the pressure was reduced by a marine regression (Supplementary Discussion).

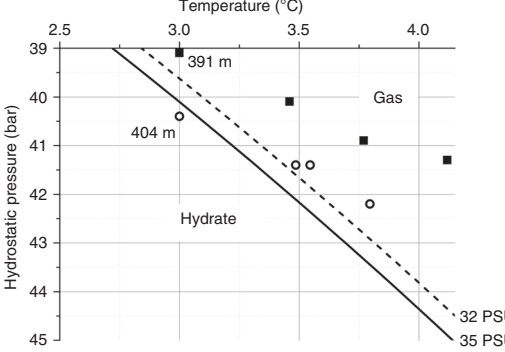

**Fig. 3** Phase boundary between free methane gas and methane hydrate. Phase boundaries are defined for structure type-I methane hydrate in sulfate-free pore water[20] for bottom water salinity (35 PSU, solid line) and the minimum salinity observed in the cores (32 PSU, broken line). In situ formation temperatures are plotted as solid squares (391 m water depth) and open circles (404 m water depth)

## Discussion

Using a transport-reaction model (details in methods section) we investigate potential scenarios of hydrate dissociation that are consistent with the geochemical variations observed within the boreholes. The 400 m deep seabed at the continental margin off northwestern Svalbard is primarily influenced by North Atlantic water[22]. The temperature of this relatively warm bottom water is highly variable and affected by the strength of the Atlantic inflow via the European Nordic Seas into the Arctic Ocean[23]. Temperature measurements conducted in the area over the last 30

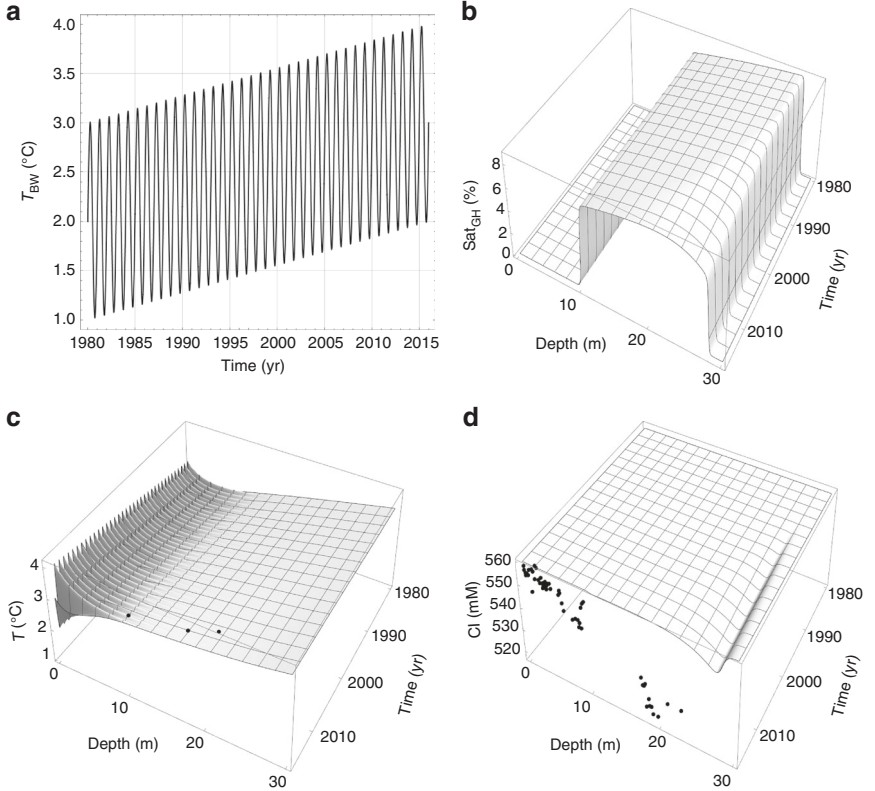

**Fig. 4** Model results for hydrate melting at 391 m water depth. **a** Bottom water temperatures ($T_{BW}$) applied as model forcing. **b** Percent of pore space occupied by gas hydrate ($Sat_{GH}$). **c** Bulk sediment temperature ($T$). Dots indicate temperatures measured in drill holes at 391 m water depth (Fig. 2). **d** Dissolved chloride concentration in pore fluids (Cl). Dots indicate concentrations in cores retrieved at 391 m water depth (Fig. 2)

years indicate mean summer values (May–October) of 2.7 °C at 400 m water depth with an interannual variability of $\pm 1$ °C[7]. Summer temperatures have increased by 1 °C over the last 30 years[7]. However, a 60-year record of bottom water summer temperatures off northwestern Svalbard at 360–400 m water depths reveals a cooling trend from 1950 to 1980 followed by a temperature rise until 2010[22]. Hence, it is unclear whether the bottom water warming observed during the last decades is due to natural variability[23] or anthropogenic forcing. Continuous temperature monitoring at 390 m water depth over a period of almost 2 years reveals strong seasonality, with minimum temperatures of 2–2.5 °C during May to June, maximum temperatures of 3.5–4 °C during November to December, and a mean annual temperature of $2.9 \pm 0.5$ °C for the year 2011[8]. Considering these observations, we conducted a series of model experiments to investigate the response of hydrates at the seabed in 391 m water depth to ambient bottom water warming.

Specifically, we model the evolution of a hydrate layer extending from 10 meters below the seafloor (mbsf) to the base of the hydrate stability zone located at 28 mbsf for an initial bottom water temperature of 2 °C and a geothermal gradient of 45 °C km$^{-1}$ (Fig. 4b). The model was forced by a linear ambient temperature increase from 2 °C in 1980 to 3 °C in 2010 superimposed over the observed seasonal cycle (Fig. 4a). Model results demonstrate that the conduction of heat through the sediment column (Fig. 4c) induced melting at the base of the hydrate stability zone as shown by the chloride depletion at 28 mbsf (Fig. 4d). However, the modeled chloride depletion is much smaller than that observed because hydrate melting in the modeled scenario is limited by slow heat conduction and mitigated by the endothermic dissociation reaction[9]. Additional model experiments conducted under alternative initial hydrate

distributions also critically fail to reproduce freshening over the scales observed in our core data (Supplementary Fig. 4). Essentially, the modeling demonstrates that more time and energy are required to yield the down-core pore water freshening. Hence, we conclude that the observed chloride depletion has not been produced by bottom water warming during the past three decades.

Surface temperatures at <200 m water depth peaked during the early Holocene (8–11 ka) throughout the Nordic seas including the area off northwestern Svalbard[24–26]. This thermal optimum was followed by slow cooling resulting in constantly low temperatures over the last few thousand years[26]. It is not known whether these surface trends also apply to bottom waters in our study area. A sediment core taken at 327 m water depth yields a trend similar to that at the surface when benthic foraminiferal $\delta^{18}O$ data are used to reconstruct ambient bottom water temperatures[26]. However, a well-calibrated benthic transfer function applied to the same core does not show the early Holocene maximum but indicates that bottom water temperatures were constant over the entire Holocene[26]. Nevertheless, we applied our model to investigate whether gas hydrate dissociation possibly induced by the early Holocene optimum might explain the observed chloride depletion (Supplementary Fig. 5). Subsurface temperatures (100–200 m) and bottom water temperatures (327 m) calculated from foraminiferal $\delta^{18}O$[26] were employed to define the model forcing. Bottom water temperatures were assumed to rise from an initial value of 2.15 °C at 13 ka to a maximum of 4.8 °C during the early Holocene. A hydrate layer extending from 16 meters below seafloor (mbsf) to 20 mbsf was assumed as the initial condition. The simulations showed that the entire layer was dissociated at 10.7 ka because of the heat that penetrated into the sediment from above. The resulting chloride minimum was

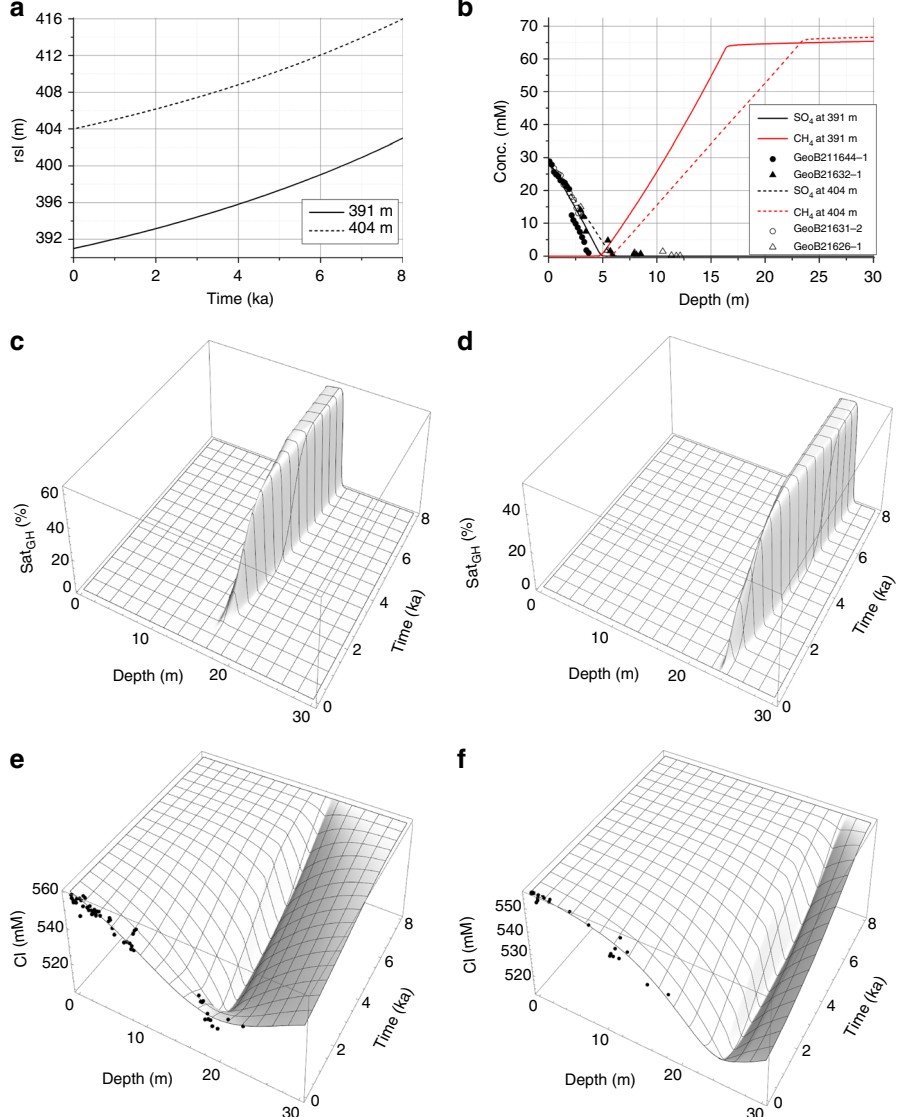

**Fig. 5** Model results for pressure-driven gas hydrate dissociation at 391 m and 404 m water depth. **a** Relative sea level (rsl) applied as model forcing. **b** Concentration of dissolved sulfate (measured: symbols; modeled: black lines) and methane (modeled: red lines) at the end of the simulation (0 ka). Shapes of modeled methane concentrations profiles match those of ex situ methane concentration profiles analyzed during the cruise (data not shown). **c** Percent of pore space occupied by gas hydrate ($Sat_{GH}$) at 391 m water depth. **d** Percent of pore space occupied by gas hydrate ($Sat_{GH}$) at 404 m water depth. **e** Dissolved chloride concentration in pore fluids (Cl) at 391 m water depth. **f** Dissolved chloride concentration in pore fluids (Cl) at 404 m water depth. Symbols indicate data in cores retrieved at 391 m and 404 m water depth

erased by molecular diffusion within a few thousand years. Hence, it is unlikely that the observed pore water anomaly was created by gas hydrate dissociation during the early Holocene.

Relative sea-level data from Prins Karls Forland[27] and northwestern Svalbard[28] clearly document a marine regression during the Holocene, with a resulting drop in hydrostatic pressure that could have induced gas hydrate dissociation. Our drill sites at the upper continental slope are located ~ 50 km west of the coastal sites where major changes in relative sea-level have been recorded[28]. The upper slope was probably not covered by a grounded glacial ice sheet. However, the northwestern rim of the ice sheet was located on the adjacent shelf break at a distance of only 5−10 km from the upper slope drill sites during Late Glacial Maximum conditions[29]. Considering the mechanical coupling between the continental shelf and upper slope, it follows that the upper slope experienced considerable isostatic depression during glacial conditions and subsequent uplift after ice sheet retreat. We use

output from an isostatically coupled ice sheet model of the retreat of the Barents Sea ice sheet[30] to constrain the postglacial rebound history in our study area on the upper continental slope off Prins Karls Forland (Supplementary Fig. 6). Relative sea-level change (Fig. 5a) was calculated from seabed uplift and eustatic sea-level[31] and applied as forcing for our sediment model to investigate whether the chloride depletion observed in the slope cores can be better explained by isostatic rebound.

Model experiments were conducted for 391 and 404 m water depth under a wide range of initial gas hydrate saturations to determine the optimal scenario depicted in Figs. 2 and 5. The experiments commence at 8 ka when the relative sea-level was 12 m higher than present and the model is forced by a decline in hydrostatic pressure determined from relative sea-level change (Fig. 5a). It is initially assumed that a gas hydrate-bearing sediment layer is present at the base of the gas hydrate stability zone (Fig. 5c, d) and that the chloride excluded during hydrate

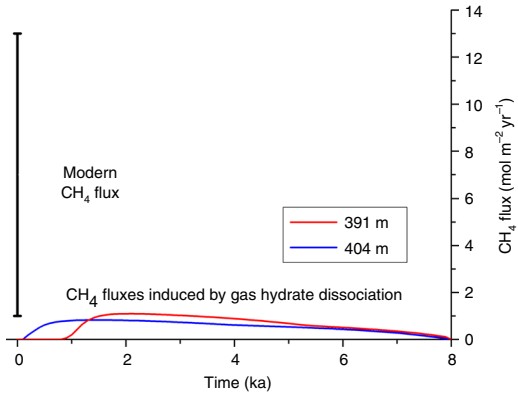

**Fig. 6** Methane gas fluxes from sediments into the overlying bottom water. Modeled fluxes induced by gas hydrate dissociation at 391 and 404 m over the last eight thousand years are compared to the area-averaged range of methane gas fluxes measured at active seeps (vertical bar) in our study area (Fig. 1)[11]

accumulation has previously diffused away. The subsequent decline in hydrostatic pressure induces an upward movement of the hydrate stability zone and hydrate dissociation at its base. Dissolved chloride concentrations decrease because dissociating hydrates release freshwater into the pore space (Fig. 5e, f). The upward displacement of the stability zone and the corresponding hydrate dissociation are mitigated by a coeval decline in salinity and temperature induced by the dissociation process itself that consumes heat and releases freshwater into the pore space[32]. Even though gas hydrates are stabilized by these negative feedback mechanisms, the dissociation front migrates upward and the hydrate layer is eroded from the bottom until the ongoing reduction of hydrostatic pressure induces complete dissociation of the hydrate layer. The resulting chloride minimum is significantly broadened over time through molecular diffusion. The final dissolved sulfate profile is controlled by upward diffusion of dissolved methane that consumes sulfate via AOM[33] (Fig. 5b). Dissolved methane in deep sediments is saturated with respect to free gas because gaseous methane fills part of the pore space initially occupied by gas hydrates at the end of the simulation.

In these model experiments, ambient bottom water temperature is maintained at a constant value of 2.5 °C until 0.1 ka when the temperature is allowed to rise exponentially to attain its modern value of 3.0 °C (Supplementary Fig. 7). Bottom water heating at the end of the simulation period is required to attain a final temperature profile consistent with the data (Fig. 2). However, we note that the heating applied over the last 100 years of the experiment resulted in no further dissociation because hydrates were already fully decomposed by seabed uplift and pressure reduction prior to this final episode of warming.

According to our transport-reaction model, most of the dissociated methane hydrate inventory (5483 mol m$^{-2}$) was released as free gas into the water column (4944 mol m$^{-2}$) at 391 m water depth. The remaining portion was dissolved in pore fluids and consumed by AOM. The calculated methane release corresponds to an annual mean flux of 0.6 mol m$^{-2}$ yr$^{-1}$ averaged over the model time-period. This gas flux should be regarded as a maximum estimate because the model does not consider the dissolution of gas bubbles in surface sediments. During the summer of 2012, a mean methane gas bubble flux of 1–13 mol m$^{-2}$ yr$^{-1}$ was measured in our study area (Fig. 1)[11]. These fluxes that are fed by a sub-seabed methane gas reservoir[8, 10] exceed the fluxes that were induced by postglacial gas hydrate dissociation by an order of magnitude (Fig. 6). We propose that gas flow from the deep reservoir would have been largely blocked in the past by the

gas hydrate layer that explains our observed chloride profiles. Within this 4 m thick layer, over 60% of the pore space was occupied by gas hydrate prior to the onset of dissociation (Fig. 5c). Such high saturations can reduce sediment permeability by up to two orders of magnitude[34, 35]. Hence, geologically derived gas fluxes into the water column are higher on the shelf and upper continental slope but decrease in deeper waters where hydrates are stable and provide a barrier to ongoing seepage[10, 36, 37]. This down-slope trend has been attributed to the sealing of permeable sediments by gas hydrate formation[36]. It has also been proposed that a portion of the gas flow is not permanently blocked but diverted up-slope until it reaches the up-dip limit of the hydrate stability zone where it seeps into the ocean[7].

Our analysis of sediment cores of Western Svalbard unambiguously confirms that retreat of the Barents Sea ice sheet led to offshore gas hydrate dissociation, a process that has been widely speculated upon from modeling and geological observations[3, 5, 38–41] but up until now, has remained unproven. Furthermore, combined modeling and geochemical analysis reveals that methane hydrates at the up-dip limit of the hydrate stability zone decomposed via postglacial isostatic rebound in contrast to previous hypotheses that invoke anthropogenic bottom water warming[7, 9]. Our data and model results also show that gas hydrates are not in themselves a significant source for gas release at the seabed. Rather, they act as a dynamic seal that blocks fluid-flow pathways for gas migration from deep geological reservoirs. Previous estimates of seafloor methane emissions by ongoing and future gas hydrate decomposition consider gas released from hydrates but ignore the potentially more significant increase in flux from underlying gas reservoirs upon hydrate dissociation[6, 23, 42]. Hence, the impact of future seabed methane fluxes on global environmental change may yet be underestimated, and further research is required to quantify the flux from deep natural gas reservoirs amplified by deterioration of the overlying hydrate seal.

## Methods

**Analytics.** Sediment samples recovered by MeBo drilling and gravity coring were transferred into the vessel's cold lab where a squeezer equipped with 0.2 µm filters was employed to separate pore fluids, applying argon pressures of 1–5 bar. Pore fluids were analyzed for chloride in the on-board laboratory applying argentometric titration as described at http://www.geomar.de/en/research/fb2/fb2-mg/benthic-biogeochemistry/mg-analytik/. Sub-samples were taken and preserved for later on-shore analyses. Ion-chromatography (IC) was employed to determine anion concentrations (Cl$^-$, SO$_4$$^{2-}$), whereas inductively coupled plasma atomic emission spectroscopy (optical ICP) was used to determine dissolved metal concentrations (lithium, boron). Dissolved chloride was determined by titration and IC. These two independent methods produced almost the same concentrations deviating in most cases by <1%. Chloride concentrations reported hereafter refer to the mean of these two measurements. IC measurements revealed that some of the pore water samples were contaminated by seawater employed as drilling fluid. Sulfate concentrations were used to correct for seawater admixture using the following two-component mixing equation:

$$C_{PW} = \frac{C_M - C_{SW} \cdot f_{SW}}{1 - f_{SW}} \qquad (1)$$

where $C_{PW}$ is the in situ pore water concentration, $C_M$ the concentration measured in samples affected by seawater admixture, $C_{SW}$ the concentration in seawater, and $f_{SW}$ the fraction of seawater in the sample. The seawater fraction ($f_{SW}$) was calculated as:

$$f_{SW} = \frac{C_{S-M}}{C_{S-SW}} \qquad (2)$$

where $C_{S-M}$ is the sulfate concentration measured in seawater-affected samples and $C_{S-SW}$ is the sulfate concentration in seawater used as drilling fluid ($C_{S-SW} = 28.93$ mM). This approach was applied to samples taken below the sulfate penetration depth only, because it assumes that the original pore water contains no sulfate. Figure 2 shows the corrected chloride concentrations. Severely contaminated samples containing more than 10 mM sulfate were discarded.

About 5 ml of wet sediment were collected at each sampled sediment depth for the analysis of sediment porosity. Porosity was determined as volume of pore water

per volume of wet sediment by weight difference before and after freeze-drying assuming a dry density of 2.5 g cm$^{-3}$ and a pore water density of 1.023 g cm$^{-3}$.

Stable oxygen and hydrogen isotope ratios ($^{18}O/^{16}O$, $^2H/^1H$) of water were analyzed by an automated equilibration unit in continuous flow mode (Gasbench 2) coupled to a Delta *plus* XP isotope ratio mass spectrometer (Thermo Fisher Scientific). Isotopic ratios are reported in δ-notation in parts per thousand (‰) relative to the VSMOW standard. Samples were measured in duplicates and the reported value is the mean value. External reproducibility based on repeated analyses of a control sample was better than 0.1 and 1‰ for δ$^{18}$O and δ$^2$H, respectively. Stable carbon isotope ratios ($^{13}C/^{12}C$) of dissolved $CH_4$ ("headspace technique") and DIC were determined by GC-isotope ratio mass spectrometry. Stable carbon isotopic ratios are reported in δ-notation in ‰ relative to the V-PDB standard (mean of at least two analytical replicates). Standard deviations of triplicate stable isotope measurements were <0.5‰.

**Geochemical modeling.** A simple numerical model was set up to evaluate the pore water data. It uses concepts developed in previous transport-reaction models[35, 43]. The model calculates fractions of bulk volume occupied by pore water, methane gas, methane hydrate, and sediment grains. It considers that gas hydrates and gas bubbles fill the pore space without supporting the grain structure such that the porosity is not affected by hydrate dissociation. Steady state compaction is considered and the resulting exponential down-core decrease in porosity is prescribed with measured porosity data. Moreover, it is assumed that the excess pressure and volume created by hydrate dissociation induces rapid gas bubble ascent and gas seepage at the sediment surface as observed in the study area. Fluid flow is ignored and gas transport is treated as a non-local process that removes gas from the sediment column directly into the overlying water column to conserve the total sediment volume and maintain hydrostatic pressure in the sediment column. Phase densities change with sediment depth but are assumed to be constant over time. The model simulates temperature, and the dissolved components chloride, sulfate and methane, the endothermic dissociation of gas hydrate into freshwater and free gas, the dissolution of gas hydrates and gas bubbles in ambient pore fluid and AOM. Dissolved chloride is an inert tracer that is transported in the water phase by molecular diffusion only, whereas dissolved sulfate and methane are consumed by AOM. Mass balance equations for the three phases hydrate, gas, and pore water are formulated as:

$$\frac{\partial \rho_H \cdot f_H}{\partial t} = -M_H \cdot (R_M + R_{HD}) \tag{3}$$

$$\frac{\partial \rho_G \cdot f_G}{\partial t} = M_G \cdot (R_M - R_{GD} - R_{EX}) \tag{4}$$

$$\frac{\partial \rho_W \cdot f_W}{\partial t} = +n_{HW} \cdot M_{H2O} \cdot R_M + M_H \cdot R_{HD} + M_G \cdot R_{GD} \tag{5}$$

where $f_i$ (i = H, G, W) are the fractions of bulk sediment volume occupied by methane hydrate (H), methane gas (G), and pore water (W), $\rho_i$ are the corresponding phase densities, $M_H$, $M_G$, and $M_{H2O}$ are the molar masses of methane hydrate ($M_H = n_{HW} M_{H2O} + M_G$), methane gas ($M_G = 16$ g mol$^{-1}$) and water ($M_{H2O} = 18$ g mol$^{-1}$), $n_{HW}$ is the number of water molecules per molecule of hydrate ($n_{HW} = 6$), $R_M$ is the molar rate of hydrate dissociation, $R_{HD}$ the rate of hydrate dissolution, $R_{GD}$ the methane gas dissolution rate, and $R_{EX}$ the rate of gas bubble expulsion.

The mass balance for dissolved chloride is formulated as:

$$\frac{\partial f_W \cdot C_{Cl}}{\partial t} = \frac{\partial}{\partial z}\left(f_W \cdot D_{Cl} \cdot \frac{\partial C_{Cl}}{\partial z}\right) \tag{6}$$

where $C_{Cl}$ is the concentration of dissolved chloride in the water phase and $D_{Cl}$ is the effective diffusion coefficient of dissolved chloride in the pore volume occupied by water. Archie's law is applied to consider the effects of tortuosity on molecular diffusion in porous media. Thus, $D_{Cl}$ is calculated as:

$$D_{Cl} = \frac{D_{MCl}}{f_W^{1-m}} \tag{7}$$

where $D_{MCl}$ is the molecular diffusion coefficients of chloride in seawater and $m$ takes a value of 2[44]. Mass balance equations for dissolved methane and sulfate are defined correspondingly:

$$\frac{\partial f_W \cdot C_{CH4}}{\partial t} = \frac{\partial}{\partial z}\left(f_W \cdot D_{CH4} \cdot \frac{\partial C_{CH4}}{\partial z}\right) + R_{GD} + R_{HD} - f_W \cdot R_{AOM} \tag{8}$$

$$\frac{\partial f_W \cdot C_{SO4}}{\partial t} = \frac{\partial}{\partial z}\left(f_W \cdot D_{SO4} \cdot \frac{\partial C_{SO4}}{\partial z}\right) - f_W \cdot R_{AOM} \tag{9}$$

where $R_{AOM}$ is the rate of anaerobic methane oxidation while $D_{CH4}$ and $D_{SO4}$ are the diffusion coefficients of methane and sulfate in pore water. The molecular diffusion coefficients are calculated as function of sediment temperature[45].

Reaction rates and concentrations of dissolved tracers are given in molar units. Concentrations and rates of anaerobic methane oxidation ($R_{AOM}$) refer to the pore water volume while the rates of hydrate dissolution ($R_{HD}$), gas bubble dissolution ($R_{GD}$), hydrate dissociation ($R_M$), and gas expulsion ($R_{EX}$) are formulated with respect to the bulk sediment volume.

The following energy equation is employed to simulate heat flow considering heat consumption during hydrate melting and multiphase conduction[43, 46]:

$$\frac{\partial}{\partial t}(C_V \cdot T) = \frac{\partial}{\partial z}\left(K_0 \cdot \frac{\partial T}{\partial z}\right) - r_T \cdot R_M \tag{10}$$

where $T$ is temperature, $C_V$ is the volumetric thermal heat capacity of the solid-water-hydrate-gas mixture, $K_0$ is the effective thermal conductivity and $r_T$ is the energy consumption during hydrate dissociation ($53.8 \times 10^3$ J mol$^{-1}$). $K_0$ and $C_V$ are defined as:

$$K_0 = K_S^{f_S} \cdot K_H^{f_H} \cdot K_W^{f_W} \cdot K_G^{f_G} \tag{11}$$

$$C_V = f_S \cdot C_S + f_H \cdot C_H + f_W \cdot C_W + f_G \cdot C_G \tag{12}$$

where the thermal conductivities and heat capacities of the individual phases are assumed to be constant over depth and time ($C_S = 0.78$ J cm$^{-3}$ K$^{-1}$, $C_W = 4.31$ J cm$^{-3}$ K$^{-1}$, $C_H = 1.82$ J cm$^{-3}$ K$^{-1}$, $C_G = 2.23$ J cm$^{-3}$ K$^{-1}$, $K_S = 1.58 \times 10^6$ J cm$^{-1}$ K$^{-1}$ yr$^{-1}$, $K_W = 1.83 \times 10^5$ J cm$^{-1}$ K$^{-1}$ yr$^{-1}$, $K_H = 1.61 \times 10^5$ J cm$^{-1}$ K$^{-1}$ yr$^{-1}$, $K_G = 1.01 \times 10^4$ J cm$^{-1}$ K$^{-1}$ yr$^{-1}$).

Molar rates of hydrate dissociation ($R_M$), gas hydrate dissolution ($R_{HD}$), and gas bubble dissolution ($R_{GD}$) are defined as[46, 47]:

$$R_M = k_M \cdot f_H \cdot \frac{\rho_H}{M_H} \cdot \text{Max}\left[1 - \frac{P_{HY}}{P_D}, 0\right] \tag{13}$$

$$R_{HD} = k_{HD} \cdot f_H \cdot \frac{\rho_H}{M_H} \cdot \text{Max}\left[1 - \frac{C_{CH4}}{C_{CH4-H}}, 0\right] \tag{14}$$

$$R_{GD} = k_{GD} \cdot f_G \cdot \frac{\rho_G}{M_G} \cdot \text{Max}\left[1 - \frac{C_{CH4}}{C_{CH4-G}}, 0\right] \tag{15}$$

where $k_M$, $k_{HD}$, and $k_{GD}$ are kinetic constants (in yr$^{-1}$), $P_D$ is the dissociation pressure of hydrate, $C_{CH4-H}$ is the concentration of dissolved methane at equilibrium with methane hydrate, and $C_{CH4-G}$ the concentration of dissolved methane at equilibrium with methane gas. According to these rate definitions, hydrates dissociate when $P_D$ exceeds the ambient hydrostatic pressure ($P_{HY}$), whereas gas hydrate and gas dissolve when the ambient concentration of dissolved methane ($C_{CH4}$) is lower than the corresponding equilibrium value. $P_D$ is calculated for each time step as a function of changing sediment temperature and pore water salinity (dissolved chloride concentration) applying a thermodynamic model[20], whereas $P_{HY}$ is continuously updated considering relative sea-level change. $C_{CH4-H}$ and $C_{CH4-G}$ are calculated as a function of sediment temperature, salinity, and hydrostatic pressure[20] while the ambient methane concentration is calculated solving the mass balance equation for dissolved methane. The kinetic constant for gas hydrate dissociation is set to a sufficiently large value ($k_M \geq 2$ yr$^{-1}$) such that the rate of endothermic hydrate dissociation is limited by heat transfer rather than the intrinsic kinetic properties of hydrate grains. The kinetic constants for hydrate and gas dissolution employed in the model ($k_{HD} \geq 1$ yr$^{-1}$, $k_{GD} \geq 1$ yr$^{-1}$) ensure that dissolved methane attains and maintains equilibrium with gas hydrate and gas in sediment layers where these phases are present.

The rate of gas expulsion ($R_{EX}$) is governed by the following equation:

$$R_{EX} = k_{EX} \cdot \frac{\rho_G}{M_G} \cdot (f_S + f_H + f_G + f_W - 1) \tag{16}$$

The kinetic constant $k_{EX}$ is set to a sufficiently large value ($\geq 1$ yr$^{-1}$) such that excess gas is expelled from the sediment and the total volume of the sediment column is conserved.

Methane is oxidized by microbial consortia using sulfate as terminal electron acceptor[48]:

$$CH_4 + SO_4^{2-} \Rightarrow HCO_3^- + HS^- + H_2O \tag{17}$$

The kinetic equation for this microbial reaction is defined as[49]:

$$R_{AOM} = k_{AOM} \cdot C_{CH4} \cdot \frac{C_{SO4}}{C_{SO4} + K_{SO4}} \tag{18}$$

where $k_{AOM}$ is a kinetic constant and $K_{SO4}$ is a Monod constant ($K_{SO4} = 1$ mM). The AOM rate is controlled by the concentration of dissolved methane, whereas dissolved sulfate is only rate-limiting when the sulfate concentration in the pore water is smaller than 1 mM[49]. The value chosen for the kinetic constant ($k_{AOM} \geq 1$ yr$^{-1}$) inhibits leakage of dissolved methane through sulfate-bearing surface sediments.

Initial gas hydrate contents were defined according to the dissolved chloride depletion observed in the pore water data, whereas initial gas saturations were set to zero. The initial temperature profile was defined applying a steady state heat flow model that considers compaction and the corresponding increase in thermal conductivity with sediment depth. Initial concentrations of dissolved chloride and sulfate were set to ambient bottom water values while methane concentrations were set to equilibrium values with respect to methane gas.

The upper boundary of the model column is located at the sediment-water interface while the lower boundary was positioned at 100 mbsf. Hydrate, gas, and water saturations and dissolved tracer concentrations were maintained at constant values at the upper and lower boundary throughout the simulation. A constant gradient, corresponding to the local geothermal gradient, was employed as lower boundary condition for temperature.

Hydrostatic pressure ($P_{HY}$) was reduced considering relative sea-level change. A corresponding $P_{HY}$ change was applied over the entire model column. Gas hydrates present in the model column were destabilized when the ambient dissociation pressure ($P_D$) exceeded the applied $P_{HY}$ value. Bottom water temperature was allowed to increase over the last 100 years of the simulation and the heat was transferred into the sediment column employing the heat flow model. The temperature increase induced a rise in $P_D$ that led to gas hydrate dissociation if ambient $P_{HY}$ was smaller than the resulting $P_D$.

The model was set up in MATHEMATICA and solved using finite differences and the method-of-lines approach as implemented in MATHEMATICA's solver for partial differential equations. The model has a resolution of 0.1 m in the top 30 m and a coarser resolution below. Mass balance calculations showed that masses and energy were conserved within an error smaller than 0.1%.

**Ice sheet modeling**. The evolution of the Barents Sea ice sheet and associated isostatic recovery of the Barents Sea continental shelf from the Last Glacial Maximum to present day is derived from a suite of model experiments carried out to reconstruct dynamics of the Eurasian ice sheet complex[30, 50, 51]. The thermo-mechanical ice model used is based on a higher-order solution to the equations governing ice sheet flow and has been verified against benchmark experiments for higher-order models[52], tested against 3D flow observations at an alpine glacier[53] and applied to a broad variety of past and present glacier and ice sheet scenarios to investigate their response to environmental and internal forcing[54–57]. The model is coupled to climate using a degree-day parameterization modified to include the effects of high latitude sublimation under extreme continental conditions[50, 58–61]. Model experiments are integrated through time on a finite-difference grid with a resolution of 10 km, with climate forcing imposed by perturbations in the NGRIP paleo isotope curve and a global eustatic sea-level curve used to determine ice flotation and calving losses at marine-terminating margins[31]. Initial ice extent, thickness and the loaded topography are inherited from a Mid-Weichselian (Marine Isotope Stage 4) experiment, allowing sufficient spin-up time for the ice sheet and isostatic loading to attain a transient equilibrium with the forcing climate at the point of kick-starting Late Weichselian experiments at 37 ka BP.

Ice thickness, extent, and the timing of advance and retreat have been constrained extensively throughout the ice complex by a diverse suite of empirical data, including geomorphological, chronological, and geophysical datasets[62–66], honoring the broad interpretations of ice sheet history inferred from the geological record. A relatively thick lithosphere of 120 km is predicted throughout the region, with a relative insensitivity to lower mantle viscosity observed at all sites[30]. Isostatic loading is calculated within the ice flow model using the commonly implemented elastic lithosphere/relaxed asthenosphere scheme[67], identified as a reasonable approach in the absence of a full spherical earth model (Supplementary Fig. 6). Relative sea-level was calculated as the difference between eustatic sealevel[31] and seabed elevation. Over 10–8 ka, the rapid rise in eustatic sea-level clearly outpaced the slow postglacial rebound (Supplementary Fig. 6). These trends were reversed after 8 ka when the global sea-level rise slowed down drastically while the seabed kept on moving upwards. Hence, relative sea-level reached a maximum at 8 ka.

**Data availability**. All relevant data are available from the authors. This includes the pore water (KW), heat flow (MR), porosity, and methane carbon isotope data (GB, TP), DIC-carbon isotope data (MET) as well as the results and code of the transport-reaction model (KW) and the results of the ice sheet model (HP). Positions, descriptions, and photographs of cores are made publicly available through the PANGAEA information system sustained by the World Data Center for Marine Environmental Sciences (WDCMARE).

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

## Acknowledgements

We would like to thank the shipboard support from the master and crew of the research vessel MARIA S. MERIAN during cruise MSM57. We are grateful to the team of the seafloor drill-rig MARUM-MeBo70. This work was partly supported by the European COST action MIGRATE, the German SUGAR program, the DFG-Research Center/ Cluster of Excellence "The Ocean in the Earth System" at Bremen University, the Cluster of Excellence "The Future Ocean" at Kiel University, the Research Council of Norway through its Centers of Excellence funding scheme, project number 223259 as well as the NORCRUST project (project number 255150) We appreciate the help from Dr. Helge Niemann for handling the O/H isotope samples.

## Author contributions

K.W., G.B., and C.B. designed the study. G.B. provided the MARUM-MeBo70 drill-rig. K.W. wrote the manuscript and developed and applied the geochemical transport-reaction model. M.R. conducted the heat measurements. K.W., W.L.H. and C.S. separated and analyzed the pore fluids. W.L.H. contributed the oxygen and hydrogen isotope data. H.P., A.H., and K.A. provided the ice sheet model and isostatic/eustatic reconstruction. T.P. provided porosity and methane carbon isotope data. C.W.H. conduct the infrared measurements. J.E.J. performed the core description. M.E.T. provided the DIC-carbon isotope data. A.H. and the other authors edited and commented on the manuscript.

## Additional information

**Competing interests:** The authors declare no competing financial interests.

