## [Peer Review File · Nature Communications]

Reviewers' comments:

Reviewer #1 (Remarks to the Author):

This is an interesting study about the role of glacial isostatic adjustment on the stability of marine gas hydrates at the continental margin off Svalbard. Previous studies had suggested that observations of gas flares were due to anthropogenic warming. The current study argues on the basis of chlorinity anomalies that hydrate decomposition occurred much earlier due to uplift of the seafloor, and that the present-day gas flares are associated with a deeper source of methane.

The arguments in the paper are fairly convincing. It is difficult to produce the chlorinity anomalies with recent temperature fluctuations. By comparison, uplift and relative sea-level change appears to be sufficient. In order to explain the geochemical observations the authors need a substantial amount of gas hydrate at the base of the stability zone (something in excess of 50% of the pore space). Presumably this amount of gas hydrate would need to be deposited during the glacial period because the prior interglacial period might be expected to cause a gas release similar to the one proposed here. I strongly urge the authors to assess the general consistency of their interpretation. Can the current rates of discharge from a deeper source achieve the required initial condition for the modeling component of this study?

Another important assertion here is that the layer of gas hydrate at the base of the stability zone acts as a barrier to gas flow. There is no doubt that filling 60% of the pore space with gas hydrate should lower the permeability, but it would be nice to know if this argument can withstand quantitative scrutiny. It seems like the nature of the sediments could also be important (one way or the other).

My final comment concerns one aspect of the thermal response of gas hydrate to recent warming. The initial concentration of gas hydrate is only 6 to 8% of the pore space, so a reader might wonder if the outcomes of the two release mechanisms are dictated by a factor of 10 difference in the initial abundance of gas hydrate. This difference should be acknowledged and discounted (if appropriate).

This is a nice study that addresses an important and topical question.

Reviewer #2 (Remarks to the Author):

The manuscript presents geochemical evidence of gas hydrate dissociation in core samples acquired on the continental slope west of Svalbard. Combined modelling of relative sea level fall and hydrate dissociation suggests that hydrate dissociation was triggered via isostatic rebound rather than by warming bottom waters.

The geographical area has been one of significant interest for methane release following the documentation of many methane gas plumes emanating from the seabed. It seems that the origin of this gas is geologic (e.g. Mau et al 2017), with a hydrate layer potentially modulating its release (e.g. Berndt et al. 2014).

The manuscript's modelled results are of interest to the wider community, suggesting as they do that isostasy related pressure reduction is the driving factor behind hydrate dissociation. The observation that one of the key implications on dissociation is the removal of a seal preventing the release of geologically derived methane is also well made.

The geochemical data presented is clear and the conclusions drawn from, in particular reduced chloride and $\delta^{13}\text{C}$, are consistent with hydrate dissociation. Supplementary material supports this case well. Modelled results concerning isostatic driven dissociation also appear sound, though this is somewhat outside my main area of expertise.

I have three main queries with the paper, as outlined below. The most significant of these is probably the third, related to potential bottom water temperature variation during the Holocene. Otherwise, the work appears rigorous and the paper is well written and cites key work well. Subject to clarification of the points raised below I suggest acceptance of the paper.

1. Elevated sulfate concentrations downcore. These are attributed to seawater intrusion. Would this not also be associated with an increase in chloride? Figure 2 doesn't seem to show this. A further clarification would be useful. Could this be redox related?

2. Previous work (e.g. Berndt et al. 2014) documented CH_4 since 3 ka BP, so a long lived emission is established, but it is unclear from the manuscript where the 8 ka date for onset of pore water freshening comes from. The only reference I find is on P.7 line 179, which simply states that the experiments commence at 8 ka. I see no reference to an Age - Depth model. Therefore the statement that dissociation commences at 8 ka needs more clarification. I assume it is based on the

commencement of rebound following ice-retreat. If so this should be clearly stated and referenced. What is the evidence that there is no emission prior to 8 ka?

3. Bottom water temperature fluctuation. The authors state (p. 5, lines 124 -125) that the bottom water temperature is variable and influenced by North Atlantic water. However, the model forced by bottom temperature increase (e.g. Figure 3) only uses a 35 year record. Whilst this is consistent with an investigation of the role of anthropogenic forcing, in my opinion more consideration should be given to the potential bottom water temperature variations during the Holocene (in the text at least). As it could be argued that water mass changes led to an increase in bottom water temperature in the early Holocene (e.g. Rasmussen et al. 2014, QSR, vol.92 pp. 280-291), which in turn led to hydrate dissociation. Some further attempt needs to be made to separate the two potential drivers (e.g. pressure and temperature).

Matt Owen

Reviewer #3 (Remarks to the Author):

Review of: Wallmann et al., Gas Hydrate Dissociation at the Continental Margin off Svalbard Induced by Isostatic Rebound rather than Global warming

review by: Peter B. Flemings

Overview:

The paper is well written and, ultimately, I think the conclusions made by the authors are valid and provide an important alternative to the view that there is hydrate warming due to anthropogenic causes in recent decades. That said, the evidence brought to bear to present the case is, to me, somewhat circumstantial and somewhat thin. I am particularly concerned about the early claim that the observed freshening is not due to hydrate dissociation during core retrieval. If the authors could defend that better, I would be more comfortable.

The major claim of the paper is that uplift and consequent pressure change caused hydrate dissociation and created a low salinity interval in the recovered core. The claims of the paper are novel and represent an important step forward from previous efforts that have focused only on the effects of thermal change in the water column. I do feel the paper will influence thinking in the field. My only concern, stated above, is that I am not convinced that the freshening is not due just to the

coring process. The authors should strengthen this justification for publication. If they can do that, I am fully supportive of publication.

1) There is a shift in the chloride concentration from 560 (I am guessing this is seawater in the area?) to about 520: the freshened water has 93 percent of the salinity of the surface water. This seems modest but real. The authors then conclude that this freshening is due to hydrate dissociation based on isotopic composition and Li and B content. I am not a pore-water geochemist, other reviewers have more depth in this field than I do. That said, it seems to be a reasonable conclusion.

2) However, what seems a stretch to me is the conclusion that we know that that hydrate dissociation was not due to the sediment coring but occurred in-situ at a geologically earlier time. The sentence on line 115 to 117 specifically states that the hydrate dissociation occurred long before coring. It implies that the supplementary information defends this. However, I think the supplementary information defends that the pore fluids are from hydrate melting. It does not, as far as I can tell, say anything about whether that melting occurred during core retrieval or long ago. The only defense of this statement is that no temperature anomaly was found in the recovered core (the reference to Trehu et al. 2004). However, as I'm sure the authors will agree, this depends completely on the time taken to recover the core, the temperature and pressure history of the core, and the initial hydrate saturation. I've recently been on an expedition where core we knew that once had hydrate had no indication of hydrate when recovered shipboard. We knew this from pressure cores of adjacent material we had taken. There might be ways you could strengthen your conclusion. For example, often we see a disturbed 'mousse-like' texture characteristic of dissociation: perhaps you could clearly state that this is not present. I would add that my guess is that the very small freshening means that the original hydrate saturation is quite small (if you assume an initial seawater salinity). Perhaps you could discuss how much hydrate was there originally if this freshening is associated with core recovery.

3) I think the modeling presented in Figure 3 is fundamentally solid. It demonstrates the interesting phenomenon that the heat has to get all the way to the deepest layer to start dissociation and that this is itself inhibited by the endothermic nature of the reaction and by the freshening. It is quite reasonable. The paper by Darnell and Flemings (2015) discusses this also. I suspect a number of other papers do as well: Darnell, K. N., and P. B. Flemings (2015), Transient seafloor venting on continental slopes from warming-induced methane hydrate dissociation, *Geophys. Res. Lett.*, 42, doi:10.1002/2015GL067012..

4) The relative sea-level change model is, I think, an effective explanation. Perhaps you state this, but I think you could emphasize that the beauty of a change in pressure is that it is transmitted immediately to every depth: there is no need for heat diffusion. In an odd way, it is like a pressure depletion hydrate production test. A minor point that I think you could illuminate more clearly is what your initial salinity assumption is. It is clear from Figure 4c that the assumption is that initially

everything is at seawater salinity. In this context, any salinity formed during hydrate formation has previously diffused away.

5) The general discussion of venting and methane volumes (p.9, 217-233) is to me reasonable. I find the statement that gas flow from the deep reservoir was largely blocked in the past to not be well defended. It seems one option is that it continued to migrate updip beneath the hydrate stability zone until it pinched out and it vented there. An alternative is that it migrated through the hydrate stability zone through a range of processes that have been previously proposed. I'm not really sure how important this statement is to your paper and it seems to me to be unnecessary hand-waving to state that flux to the ocean was reduced in the past.

6) I find the statement 'our down-core analysis unambiguously confirms that the Barents Sea ice sheet retreat led to offshore gas hydrate dissociation' to be a little strong (line 239) due to my questions about whether this could be associated with core retrieval.

7) Again, I find the argument on p.10 that hydrates are acting as a seal to venting to not be substantiated by your data or your model (which is 1d). To me, the obvious situation is that the gas just migrates a little further updip beneath the hydrate layer and vents where the hydrate layer thins to zero. Thus the argument that the hydrate is a regulator of an underlying gas flux is weak.

8) I broadly think your numerical model is solid (p.11). That said, I think it is worth pointing out that other models do suggest re-formation of methane hydrate above where it is melted (again Darnell and Flemings, 2016) whereas you just vent it through the hydrate stability zone. To be honest, I'm not sure this makes much of a difference to your fundamental conclusion. I would argue that your model is a maximum venting case and your whole point is that this is not very much.

Reply to the reviewers' comments

We would like to thank the reviewers for their critical, helpful and very constructive comments that forced us to reconsider our conclusions and motivated us to add new material to support our claims. We think that our response to the reviews led to a further improvement of the manuscript.

Reviewer #1

Comment: *“Can the current rates of discharge from a deeper source achieve the required initial condition for the modeling component of this study?”*

Reply: The modern rate of methane gas leakage at our 391 m study area has been determined as $1 - 13 \text{ mol m}^{-2} \text{ yr}^{-1}$. This flux probably represents the gas flux from a deeper source that was responsible for the built-up of the gas hydrate layer that we apply as initial condition. We assumed that 5483 mol m^{-2} of methane accumulated in the sediment as gas hydrate prior to the onset of dissociation. Hence, it may have taken 400 - 5500 years of continuous gas ascent to accumulate this amount of gas hydrate. This seems to be a plausible accumulation period for our geological setting.

Comment: *“There is no doubt that filling 60% of the pore space with gas hydrate should lower the permeability, but it would be nice to know if this argument can withstand quantitative scrutiny.”*

Reply: The most widely used sediment model assumes that gas hydrate is formed in the center of cylindrical pores. For this simple scenario, the decrease in intrinsic permeability (k) induced by gas hydrate formation can be estimated as^{2,3}:

$$k = k_0 \cdot \left(1 - S_h^2 + \frac{2 \cdot (1 - S_h)^2}{\ln(S_h)} \right)$$

where k_0 is permeability in the absence of hydrate and S_h is the fraction of pore space filled by gas hydrate. This equation indicates that permeability is reduced by two orders of magnitude when 60 % of the pore space is filled by gas hydrate ($k/k_0 \approx 0.01$ for $S_h = 0.6$). We introduced the following sentence in the revised manuscript to give a quantitative estimate of the effect:

Lines 253-254 (p. 9): “Such high saturations may reduce sediment permeability by about two orders of magnitude”.

Comment: *“My final comment concerns one aspect of the thermal response of gas hydrate to recent warming. The initial concentration of gas hydrate is only 6 to 8% of the pore space, so a reader might wonder if the outcomes of the two release mechanisms are dictated by a factor of 10 difference in the initial abundance of gas hydrate. This difference should be acknowledged and discounted (if appropriate).”*

Reply: Additional model runs are presented in the supplementary information. One of these simulates the dissociation of a hydrate layer with an initial saturation of 25 % (last row of Fig. S4). In this case, the decrease in saturation induced by bottom water warming is so small relative to the high initial value that it is barely resolved in the plot. We decided to show a simulation with a smaller

initial saturation in the main text such that the reader is able to actually see the small changes in hydrate saturation induced by 30 years of bottom water warming.

Reviewer #2 (Matt Owen)

Comment: "Elevated sulfate concentrations downcore. These are attributed to seawater intrusion. Would this not also be associated with an increase in chloride? Figure 2 doesn't seem to show this. A further clarification would be useful. Could this be redox related?"

Reply: Fig. 2 shows corrected chloride concentrations. They were corrected as described in the methods section to account for the intrusion of seawater. We added the following sentence to the legend of Fig. 2 to clarify this point:

Lines 113-114 (p. 4): "Chloride concentrations were corrected to account for seawater intrusion during the drilling process (methods section)."

There is no indication for redox changes in the sulfate-bearing sediment section. We know from previous MeBo deployments that the drilling fluid (ambient bottom water) penetrates preferentially into sandy sediment layers due their high permeability. This was apparently also the case during our deployments off Svalbard. We registered sulfate contaminations in sandy layers while clay-rich layers were not contaminated.

Comment: "...the statement that dissociation commences at 8 ka needs more clarification. I assume it is based on the commencement of rebound following ice-retreat. If so this should be clearly stated and referenced. What is the evidence that there is no emission prior to 8 ka?"

Reply: The relative sea-level (rsl) reached a maximum at 8 ka and decrease over the following period as shown in Fig. 5a. To better explain why our simulations start at 8 ka, we expanded the scale of Fig. S6 and added the following text to the supplementary information:

Lines 141-144 (supp.p. 7): "Relative sea-level was calculated as the difference between eustatic sealevel⁴ and seabed elevation. Over 10 – 8 ka, the rapid rise in eustatic sea-level clearly outpaced the slow post-glacial rebound (Fig. S6). These trends were reversed after 8 ka when the global sea-level rise slowed down drastically while the seabed kept on moving upwards. Hence, relative sea-level reached a maximum at 8 ka."

It is difficult to reconstruct what happened prior to 8 ka. According to our isostatic rebound model, rsl rose from 10 ka to 8 ka. If the bottom water temperature was stable over this period, the marine transgression may have induced an expansion of the gas hydrate stability zone and the built up of the gas hydrate inventory that dissociated subsequently at < 8ka.

Comment: "...more consideration should be given to the potential bottom water temperature variations during the Holocene (in the text at least). As it could be argued that water mass changes led to an increase in bottom water temperature in the early Holocene (e.g. Rasmussen et al. 2014, QSR, vol.92 pp. 280-291), which in turn led to hydrate dissociation. Some further attempt needs to be made to separate the two potential drivers (e.g. pressure and temperature)."

Reply: We added a new model run simulating dissociation of gas hydrates induced by early Holocene warming to the supplementary information (Fig. S5).

Fig. S5. Model results for hydrate melting at 391 m water depth. A. Bottom water temperatures (T_{BW}) applied as model forcing. B. Percent of pore space occupied by gas hydrate (Sat_{GH}). C. Bulk sediment temperature (T). Dots indicate temperatures measured in drill holes at 391 m water depth (s. Fig. 2). D. Dissolved chloride concentration in pore fluids (Cl). Dots indicate concentrations in cores retrieved at 391 m water depth (s. Fig. 2).

The new figure is accompanied by the following caption and text:

Lines 95 – 106 (supp. p. 5):

“Gas hydrate dissociation induced by early Holocene warming

An additional model run was executed to simulate gas hydrate dissociation induced by bottom water warming over the early Holocene. Subsurface temperatures (100 – 200 m) and bottom water temperatures (327 m) calculated from foraminiferal $\delta^{18}\text{O}^5$ were employed to define the model forcing (Fig. S5A). Bottom water temperatures were assumed to rise from an initial value of 2.15°C at 13 ka to a maximum of 4.8 °C during the early Holocene (Fig. S5A). A hydrate layer extending from 16 meters below seafloor (mbsf) to 20 mbsf was assumed as the initial condition (Fig. S5B). The simulations showed that the entire layer was dissociated at 10.7 ka (Fig. S5B) because of the heat that penetrated into the sediment from above (Fig. S5C). The resulting chloride minimum was erased by molecular diffusion within a few thousand years (Fig. S5D). Hence, dissociation of gas hydrate over the early Holocene cannot produce significant chloride depletion in modern pore fluids (Fig. S5D).”

Moreover, we added the following text to the Discussion section of the main text:

Lines 174 – 187 (p. 6 -7): “Surface temperatures at < 200 m water depth peaked during the early Holocene (8 – 11 ka) throughout the Nordic seas including the area off northwestern Svalbard^{5,6,7}. This thermal optimum was followed by slow cooling resulting in constantly low temperatures over the last few thousand years⁵. It is not known whether these surface trends also apply to bottom waters in our study area. A sediment core taken at 327 m water depth yields a trend similar to that at the surface when benthic foraminiferal $\delta^{18}\text{O}$ data are used to reconstruct ambient bottom water temperatures⁵. However, a well calibrated benthic transfer function applied to the same core does not show the early Holocene maximum but indicates that bottom water temperatures were constant over the entire Holocene⁵. Nevertheless, we applied our model to investigate whether gas hydrate dissociation possibly induced by the early Holocene optimum might explain the observed chloride depletion (supplementary information). The simulation clearly shows that the salinity minimum induced by dissociation is not conserved but almost completely erased by molecular diffusion over the Holocene (Fig. S5). Hence, it is unlikely that the observed pore water anomaly was created by gas hydrate dissociation during the early Holocene.”

Reviewer #3 (Peter B. Flemings)

Comment: “I am particularly concerned about the early claim that the observed freshening is not due to hydrate dissociation during core retrieval. If the authors could defend that better, I would be more comfortable. The only defense of this statement is that no temperature anomaly was found in the recovered core (the reference to Trehu et al. 2004). However, as I’m sure the authors will agree, this depends completely on the time taken to recover the core, the temperature and pressure history of the core, and the initial hydrate saturation. I’ve recently been on an expedition where core we knew that once had hydrate had no indication of hydrate when recovered shipboard. Often we see a disturbed ‘mousse-like’ texture characteristic of dissociation: perhaps you could clearly state that this is not present”

Reply: We added a new figure to the results section (Fig. 3, page 4)

Fig. 3. Phase boundary between free methane gas and structure type-I methane hydrate in sulfate-free pore water⁸ for bottom water salinity (35 PSU, solid line) and the minimum salinity observed in the cores (32 PSU, broken line). In-situ formation temperatures are plotted as solid squares (391 m water depth) and open circles (404 m water depth).

and the following text:

Lines (123-125, p. 5): “The in-situ temperature measurements clearly show that methane hydrate was not stable in the cores taken at 391 m water depth while at 404 m only the uppermost sediment section was located within the gas hydrate stability zone during the time of sampling (Fig. 3).”

We think that this important observation largely excludes the possibility that hydrates were present in the core and produced the chloride depletion upon core retrieval. We did not observe a mousse-like texture in our cores. Moreover, water and air temperatures were much lower offshore Svalbard than in the Gulf of Mexico and we performed our IR measurements within one hour after core retrieval. Thus, we are confident that we would have picked up an IR cooling signal if hydrate were present in the cores.

Comment: “A minor point that I think you could illuminate more clearly is what your initial salinity assumption is. It is clear from Figure 4c that the assumption is that initially everything is at seawater salinity. In this context, any salinity formed during hydrate formation has previously diffused away.”

We added the following sentence to the Discussion:

Lines 208 -209 (p.7): “and that the chloride excluded during hydrate accumulation has previously diffused away.”

Comment: “I find the statement that gas flow from the deep reservoir was largely blocked in the past to not be well defended. It seems one option is that it continued to migrate updip beneath the hydrate stability zone until it pinched out and it vented there. An alternative is that it migrated through the hydrate stability zone through a range of processes that have been previously proposed. I’m not really sure how important this statement is to your paper and it seems to me to be unnecessary hand-waving to state that flux to the ocean was reduced in the past. To me, the obvious situation is that the gas just migrates a little further updip beneath the hydrate layer and vents where the hydrate layer thins to zero.”

Reply: We added the following sentence to the Discussion section to account for this possibility:

Lines 257-259 (p. 9). “It has also been proposed that a portion of the gas flow is not permanently blocked but diverted up-slope until it reaches the up-dip limit of the hydrate stability zone where it seeps into the ocean⁹.”

However, in many cases gas flow is only possible through distinct high-permeability pathways. The gas is kept in the subsurface when these pathways are clogged by gas hydrate formation. Hence, we maintain our conclusion that gas seepage from deep geological reservoirs may increase upon hydrate dissociation.

Comment: “I broadly think your numerical model is solid (p.11). That said, I think it is worth pointing out that other models do suggest re-formation of methane hydrate above where it is melted (again Darnell and Flemings, 2016) whereas you just vent it through the hydrate stability zone. To be honest, I’m not sure this makes much of a difference to your fundamental conclusion. I would argue that your model is a maximum venting case and your whole point is that this is not very much.”

We agree with this statement. On lines 246-247 we actually wrote: “This gas flux should be regarded as a maximum estimate because the model does not consider the dissolution of gas bubbles in surface sediments.”

References

1. Sahling H, Römer M, Pape T, Berges B, Fereirra CD, Boelmann J, *et al.* Gas emissions at the continental margin west of Svalbard: mapping, sampling, and quantification. *Biogeosciences* 2014, **11**(21): 6029-6046.
2. Kleinberg RL, Flaum C, Griffin DD, Brewer PG, Malby GE, Peltzer ET, *et al.* Deep sea NMR: Methane hydrate growth habit in porous media and its relationship to hydraulic permeability, deposit accumulation, and submarine slope stability. *Journal of Geophysical Research-Solid Earth* 2003, **108**(B10): 17.
3. Liu XL, Flemings PB. Dynamic multiphase flow model of hydrate formation in marine sediments. *Journal of Geophysical Research-Solid Earth* 2007, **112**(B3): 23.
4. Waelbroeck C, Labeyrie L, Michel E, Duplessy J-C, McManus JF, Lambeck K, *et al.* Sea-level and deep water temperature changes derived from benthic foraminifera isotopic records. *Quaternary Science Reviews* 2002, **21**: 295-305.
5. Rasmussen TL, Thomsen E, Skirbekk K, Slubowska-Woldengen M, Kristensen DK, Koc N. Spatial and temporal distribution of Holocene temperature maxima in the northern Nordic seas: interplay of Atlantic-, Arctic- and polar water masses. *Quaternary Science Reviews* 2014, **92**: 280-291.
6. Sarnthein M, Van Kreveld S, Erlenkeuser H, Grootes PM, Kucera M, Pflaumann U, *et al.* Centennial-to-millennial-scale periodicities of Holocene climate and sediment injections off the western Barents shelf, 75 degrees N. *Boreas* 2003, **32**(3): 447-461.

7. Hald M, Andersson C, Ebbesen H, Jansen E, Klitgaard-Kristensen D, Risebrobakken L, *et al.* Variations in temperature and extent of Atlantic Water in the northern North Atlantic during the Holocene. *Quaternary Science Reviews* 2007, **26**(25-28): 3423-3440.
8. Tishchenko P, Hensen C, Wallmann K, Wong CS. Calculation of the stability and solubility of methane hydrate in seawater. *Chemical Geology* 2005, **219**: 37-52.
9. Westbrook GK, Thatcher KE, Rohling EJ, Piotrowski AM, Pälike H, Osborne AH, *et al.* Escape of methane gas from the seabed along the West Spitsbergen continental margin. *Geophysical Research Letters* 2009, **36**(L15608): doi:10.1029/2009GL039191.

REVIEWERS' COMMENTS:

Reviewer #1 (Remarks to the Author):

The authors have responded to my comments on the original draft. The current version is suitable for publication.

Reviewer #2 (Remarks to the Author):

The revised manuscript has addressed the concerns raised with the initial submission.

In particular, the model run investigating the early Holocene optimum is a welcome addition, strengthening the case in favour of isostatic rebound rather than temperature change.

On the basis of the revisions made I recommend acceptance of the manuscript.

Reviewer #3 (Remarks to the Author):

I have previously reviewed this manuscript. I find the edits and additions to fully meet my expectations. I look forwards to seeing the publication.

REVIEWERS' COMMENTS:

Reviewer #1 (Remarks to the Author):

The authors have responded to my comments on the original draft. The current version is suitable for publication.

Reviewer #2 (Remarks to the Author):

The revised manuscript has addressed the concerns raised with the initial submission.

In particular, the model run investigating the early Holocene optimum is a welcome addition, strengthening the case in favour of isostatic rebound rather than temperature change.

On the basis of the revisions made I recommend acceptance of the manuscript.

Reviewer #3 (Remarks to the Author):

I have previously reviewed this manuscript. I find the edits and additions to fully meet my expectations. I look forwards to seeing the publication.

Reply to the reviewer's comments

Thank you very much for your valuable time and very helpful comments on our initial draft.